# Metal Content, Fatty Acid and Vitamins in Commercially Available Canned Fish on the Bulgarian Market: Benefit–Risk Ratio Intake

**DOI:** 10.3390/foods13060936

**Published:** 2024-03-19

**Authors:** Katya Peycheva, Veselina Panayotova, Tatyana Hristova, Albena Merdzhanova, Diana Dobreva, Tonika Stoycheva, Rositsa Stancheva, Patrizia Licata, Francesco Fazio

**Affiliations:** 1Department of Chemistry, Medical University of Varna, 9002 Varna, Bulgaria; veselina.ivanova@hotmail.com (V.P.); hristova_tatyana@yahoo.com (T.H.); a.merdzhanova@gmail.com (A.M.); diana@mu-varna.bg (D.D.); tonika.vladislavova@mu-varna.bg (T.S.); rositsa.stancheva@mu-varna.bg (R.S.); 2Department of Veterinary Sciences, University of Messina, Polo SS Annunziata, 98168 Messina, Italy; plicata@unime.it (P.L.); francesco.fazio@unime.it (F.F.)

**Keywords:** canned fish, toxic and essential elements, fatty acids, vitamins, health risk

## Abstract

Today, the distribution and consumption of canned products have become widespread due to the convenience of using processed fish products. This study aims to evaluate elements of toxicological concern and essential elements (Cd, Al, Cu, Cr, Mn, Fe, Pb, Ni, and Zn), the fatty acid composition, and the fat-soluble vitamin and antioxidant pigment contents of various canned fish products purchased on the Bulgarian market. The estimated weekly intake and human health benefit–risk ratio based on metal elements and n-3 LC-PUFA contents in canned fish products were assessed. The contents of the analyzed elements in the canned samples were found to be below the limits set by various health organizations. Al was detected in only one sample. The profile of fatty acids showed that the canned fish had high PUFA/SFA ratios, EPA + DHA contents and low SFA, AI and TI values. The analyzed canned fish contained significant contents of fat-soluble vitamins. The Atlantic bonito in a jar sample was characterized by a high percentage of vitamin A (26.7% RDI) and vitamin D3 (142% RDI) per 100 g portion. The calculated EWI value shows that the consumption of canned fish products did not present any risk. The benefit–risk ratio indicates that the canned fish species are safe for human consumption, except for one sample regarding Cr.

## 1. Introduction

Seafood is very rich in omega-3 long-chain polyunsaturated fatty acids (such as n-3 LC-PUFAs) and is an important source of high-quality animal protein [1,2,3] and bioactive compounds such as carotenoids [3] and vitamins [4,5], with excellent human health properties [6,7]. An alternative way to introduce seafood into the normal diet is canning (fish or fishery products), which is popular and widely practiced in developed countries, since canned products are convenient, ready-to-eat and affordable [8,9]. The canning process allows food to be safely distributed, stored, and consumed over an extended period on a global scale. The canning process ensures preservation and the delayed use of fish and fish products. The shelf life of canned fish ranges from 1 to 5 years.

The canning of marine species is responsible for about 9.5–11.5% of the total world fishery production according to the FAO database for the period 2010–2019, corresponding to around 16.3–17.5 million tons per year [10]. Among the different species, the most preferred species for use in canning mass production worldwide is tuna, followed by anchovies, salmon, sardines, mackerels and herring [8,9,11,12]. Fish is commonly packed in either oil or brine. The most common types of oil used in canning are olive oil and refined seed oils. 

In addition to their beneficial health effects, canned fish products may pose several risks, leading to adverse consumer impacts. For example, contamination can be observed during the processing process, during storage or even in the fish habitat areas [8,9]. Therefore, it is of importance to regularly monitor canned fish products offered to consumers regarding various pollutants [9]. 

Canned fish which is produced in Bulgaria is mainly offered to local consumers and is exported to the global market. However, data on the contents of trace elements, fatty acids, fat-soluble vitamins, antioxidant pigments and cholesterol in canned fish produced in Bulgaria are very limited and local expert authorities such as the Bulgarian Ministry of Health and National Center for Public Health and Analysis stated that canned fish should be analyzed more frequently with respect to some biologically active components and contaminants.

Our study covers various canned fish products, which were selected as they are ready to consume, microbiologically safe, a preferred nutritional consumer choice and are experiencing increased economic demand in Bulgaria. Shelf-stable fish products account for almost 32% of all fish products in our country [13]. Therefore, this study investigated (1) the concentration of some elements of toxicological concern (Cd, Pb and Al) and essential elements (Cu, Cr, Mn, Ni, Fe, Zn), the fatty acid composition and the fat-soluble vitamin, antioxidant pigment and cholesterol contents of canned fish products purchased from the Bulgarian grocery market during 2023; (2) the impact of canned fish products on human health using a commonly used risk index based on the maximum values of elements that could be reached via the consumption of canned fish products; and (3) the benefit–risk ratio for consumer’s health based on essential element and n-3 LC-PUFA concentration in canned fish products.

## 2. Materials and Methods

### 2.1. Canned Fish Collecting

Samples were obtained randomly from supermarkets and grocery shops in the region of Varna and Burgas, Bulgaria, during the period from May 2023 to August 2023. Seven canned fish samples (in brine, in olive oil, and with spices) from five different companies were used in this study. Table 1 presents the types of fish, packaging information, quantities and other information such as the additives and types of sauce used by the canned fish companies.

A minimum of three samples were selected from each of the seven canned fish products to provide a representative dataset. The analyzed canned fish samples were locally produced. After transportation to the laboratory, the cans were opened, and the liquid content was removed by straining the sample for 20 min using a plastic sieve with size of 2 mm. The fish flesh was homogenized thoroughly using a food processor. Samples were then digested immediately and analyzed in respect to selected elements of toxicological concern (Cd, Pb and Al), essential elements (Fe, Cr, Mn, Ni, Zn, Cu), fatty acid composition and fat-soluble vitamin, antioxidant pigment and cholesterol content.

### 2.2. Analytical Determinations

#### 2.2.1. Determination of Toxic and Essential Elements

The Milli-Q water system (Millipore, Bedford, MA, USA) was used to obtain analytical-grade water, which was further used for reagent and standard solution preparation and dilution. All reagents in this study were also of analytical reagent grade. The laboratory glassware was washed in diluted HNO_3_ and rinsed with Milli-Q water before use. Then, 8 cm^3^ of 65% (*w*/*v*) HNO_3_ (Merck, Darmstadt, Germany) and 2 cm^3^ of 30% (*w*/*v*) H_2_O_2_ (Fisher Scientific, Leicestershire, UK) were used for acid digestion. A ~1 g wet weight tissue sample was weighed and then placed in Teflon digestion vessels and subjected to digestion by the microwave closed-vessel digestion system MARS 6 (CEM Corporation, Matthews, NC, USA) (3-stage program; maximum temperature, 210 °C; maximum power, 1050 W; maximum pressure, 800 psi). After complete dissolution, the fish samples were cooled to 30 °C, diluted to the required volume (25 mL) and kept in polyethylene bottles prior to analysis.

The trace element calibration standard solutions were prepared by diluting an ICP multi-element standard solution IV (1000 mg/L in 2% HNO_3_) purchased from Sigma Aldrich. For the determination of the concentrations of Cd, Al, Cu, Cr, Zn, Fe, Pb, Mn and Ni, an ICP OES Spectrometer (Optima 8000, Perkin Elmer, Waltham, MA, USA) was used. The operating parameters of ICP OES were as follows: plasma gas flow—8 L/min; nebulizer gas flow—0.6 L/min; auxiliary gas flow—0.4 L/min; RF power—1500 watts; axial plasma view; peristatic pump flow rate—1.5 mL/min; processing peak, area/height; read parameters auto (1 to 5 (min–max)); spray chamber, cyclonic glass; nebulizer, Concentric Glass, MEINHARD^®^. The accuracy of metal analyses was ensured by using DORM-2 (NRCC, Ottawa, ON, Canada) certified dogfish tissue as the calibration standard. Good recoveries (between 90.5 and 108%) were observed. The CRM was digested and analyzed in the same way as the analytical samples.

The analyses were carried out in triplicate, and the significance level was chosen as 0.05.

#### 2.2.2. Determination of Fatty Acids

Lipids were extracted according to the Bligh and Dyer method [14] with some modifications. Around 3 g of tissue homogenates were extracted sequentially with CHCl_3_/CH_3_OH (1:2 *v*/*v*), CHCl_3_/CH_3_OH (1:1 *v*/*v*), and CHCl_3_ with constant mixing for 30 min after each extraction step. Phase separation was completed with NaCl solution in H_2_O (0.9% *w*/*v*). After centrifugation (3500× *g*, 15 min), the bottom chloroform layer was collected using a Pasteur pipette, filtered through Na_2_SO_4_ and the solvent evaporated to dryness by means of a rotary evaporator. The total lipid content was determined gravimetrically. The extracted lipids were diluted (up 1 mL) with hexane and frozen and kept at −18 °C for further analysis. 

The fatty acid (FA) composition was determined as mainly comprising fatty acid methyl esters (FAMEs) after acid catalyzed transesterification with 1 mL 5% H_2_SO_4_–CH_3_OH in a 70 °C water bath [15]. FAMEs were analyzed using gas chromatography mass spectrometry, Thermo Fisher Scientific FOCUS—PolarisQ Ion Trap GC-MS (Thermo Fisher Scientific, Waltham, MA, USA). An extract of 1 μL, containing the total FAMEs, was injected into a capillary column (Trace™ TR-FAME, 60 m × 0.25 mm × 0.25 μm) with a split ratio of 10:1. The carrier gas was helium at a flow rate of 1.2 mL/min and a temperature program as follows: an initial oven temperature of 100 °C for 1 min, followed by an increase at a rate of 10 °C/min from 100 °C to 160 °C, an increase at a rate of 5 °C/min from 160 °C to 215 °C, which was held for 6 min, and next an increase at a rate of 5 °C/min from 215 °C to 230 °C, which was held for 5 min. FAME peaks were identified based on the comparison of the retention times with the authentic standards (Supelco 37 Component FAME Mix and PUFA № 3 from Menhaden oil). The results were presented as weight % of total fatty acids.

#### 2.2.3. Determination of Fat-Soluble Vitamins, Antioxidant Pigments and Cholesterol

The qualitative and quantitative determination of fat-soluble vitamins (A, D_3_ and E), antioxidant pigments (β-carotene and astaxanthin) and cholesterol were performed simultaneously by means of liquid chromatography analysis. The HPLC system (Thermo Scientific Spectra SYSTEM) was equipped with a reversed-phase chromatography column (Synergi Hydro-RP 80A 4 μ 250 × 4.6 mm). The analyte extraction method involved alkaline hydrolysis and subsequent liquid–liquid extraction with a non-polar solvent. Chromatography conditions and sample preparation were described in detail by Dobreva et al. [16].

### 2.3. Consumer Health Risk Estimation

The values of estimated weekly intake (EWI) and benefit–risk ratio (BRR) were calculated to assess the human health risks posed to consumers related to the intake of canned fish samples. The calculations for EWI were based on the acceptance that individuals consumed 0.190 kg/person/week of canned fish, provided by the National Statistical Institute in Bulgaria [17], and taking into account that there is no detailed information about the type of canned fish consumption in our country. 

#### 2.3.1. Determination of Heavy and Essential Elements

The consumption of canned fish will vary greatly from one person to another [8]. The consumption of various canned fish products was evaluated on the basis of estimated weekly intake (EWI) using the formula [18,19,20]
$$EWI=\frac{\left(M_{c}\times FIR\right)}{B_{w}}$$
where *EWI* is the estimated weekly intake (mg/kg bw/week), *M_c_* is the average concentration of the analyzed element (mg/kg), *F_IR_* is the average weekly consumption rate (kg/person), and *B_w_* is the consumer body weight (kg). In this study, 70 kg was nominally used as the average body weight for adults (18–25 age group). 

The calculated EWI values and the provisional tolerable weekly intake (PTWI) levels established by the FAO/WHO Joint Expert Committee on Food Additives (JECFA) and/or the European Food Safety Authority (EFSA) were compared. 

#### 2.3.2. Benefit–Risk Ratio

The hazard quotient for BRR was estimated using the equation proposed by Gladyshev et al. [21]:$${HQ}_{EFA}=\frac{R_{EFA}\times C_{element}}{C\times RfD\times B_{w}}$$
where *R_EFA_* is the recommended daily dose of EPA+DHA for a person (mg/day), *C_element_* is the concentration of the elements of toxicological concern/essential elements (mg/kg), *C* is the content of EPA+DHA in a given fish (mg/g), *RfD* is the reference dose (μg/kg/d), and *B_w_* is the average adult body weight (70 kg). An HQ_EFA_ < 1 indicates benefits for human health from fish consumption, while values of HQ_EFA_ > 1 are associated with higher risk [21]. For the calculation of this equation, the recommended daily dose of EPA + DHA is 500 mg/day [22,23], while *RfD* values were obtained from the EPA Region III Risk-Based Concentrations summary table [24], excluding Pb [25].

### 2.4. Statistical Analysis

The Microsoft Office Excel 2010 software with significance at *p* < 0.05 was used for the descriptive statistics and one-way analysis of variance (ANOVA).

## 3. Results and Discussion

### 3.1. Heavy Metals Content of Canned Fish Samples

The total concentration of the analyzed metals in the samples is presented in Table 2. The lowest Zn contents were found in canned Atlantic bonito (0.93 ± 0.14 mg/kg ww), while the highest Zn contents were observed in canned European sprat (4.37 ± 0.39 mg/kg ww), although none of the samples exceeded the permissible limit (50 mg/kg) according to Bulgarian Food Codex for marine fishes [26]. The data in the literature show values between 11.605 mg/kg (canned rainbow trout) and 22.467 mg/kg (canned anchovies) for Turkey [12], with similar results reported by Tuzen and Soylak [27]; the average zinc content in canned tuna collected from supermarkets in the region of Vojvodina, Serbia was 21.96 mg kg^−1^, while in canned sardines it was 18.21 mg kg^−1^ and in canned smoked sprouts it was 21.53 mg kg^−1^ [28].

Copper is essential in promoting a good health condition, but very high intake can cause negative health problems such as kidney damage and liver failure [29,30]. Therefore, the Bulgarian Food Codex [26] recommended a limit for Cu in marine fishes of 10 mg/kg; FAO (1983) [31] and Maff (1995) [32] recommended a limit of 30 mg/kg. In our study, the average Cu concentration was as high as 0.48 mg/kg for the canned sample of bluefish in a metal can. In the literature, the mean Cu concentrations were reported to be 1.77 mg/kg in canned anchovy [33], 0.32 mg/kg in canned pink salmon, and 0.47 mg/kg in canned red salmon [34], 1.145 mg/kg in canned anchovies and 0.541 mg/kg in rainbow trout [12], 2.50 μg/g in canned tuna fish and 1.10 μg/g in canned *Trachurus trachurus* obtained from supermarkets in Turkey in 2005 [27].

The iron level in our canned fish samples are similar to the values reported by other authors [9,12,27,28,35]. European Union legislation [36] has not established maximum values for Fe, but Serbian regulations have set a maximum limit for Fe in canned fishery products as 30 mg/kg, while in Turkey, the recommended maximum limit for Fe in canned foods (including fish and fishery products) is 15 mg/kg [12]. The higher value of Fe in the samples may be caused by pollution of the raw material, the environment itself or other contamination sources (containers, poor handling practices of raw materials, or processing steps on land and/or at sea) [12,37]. 

Among the samples, those with the highest Mn content are those from the European sprat (0.23 mg/kg ww). The minimum and maximum manganese values in the literature were found to be 0.90 μg/g in canned tuna fish and 2.50 μg/g in canned anchovy fish samples produced and marketed in Turkey in 2005 [27], and 0.012 mg/kg for canned tuna fish, 0.115 mg/kg for canned chub mackerel and 0.524 mg/kg for canned whelk in commonly consumed canned marine products in South Korea [38]. According to FAO/WHO, the recommended maximum concentration of Mn in food should not exceed 3.00 mg/kg [39].

The canned fish product with the highest Cr values was Atlantic bonito in vegetable oil (0.54 ± 0.01 mg/kg). The values of Cr values found by other authors vary significantly: from 0.06 to 4.08 mg/kg ww in 34 canned fish samples from seven companies obtained from local markets in Turkey in 2021 [9]; from 0.97 μg/g in canned sardine to 1.70 μg/g in canned anchovy fish samples in a Turkish study by Tuzen and Soylak (2007) [27]; and between 0.020 μg/g and 0.038 μg/g in canned tuna and sardine commercially available in the Latin American market [35]. The Bulgarian Food Codex [26] reported maximum limits for Cr in marine fishes of 0.3 mg/kg f.w. Additionally, Cr treatment is usually employed to make the Sn layer of cans less vulnerable to environmental impurities and increase the coating adherence [35,40]. 

Ni is of great importance to be determined in foods, since it has a potential to migrate during food processing or packaging [35,41]. Usually, canned fish fillets are boiled in Ni-containing vessels, from which contamination could appear [42]. The content of Ni in the canned fish samples collected in our study varied between 0.011 and 0.065 mg/kg ww. The content of Ni has been reported to be between 0.045 μg/g ww. for canned tuna and 0.056 μg/g ww. for canned sardines available commercially on the markets of Latin America [35]; between 0.42 μg/g in canned Black Sea bonito and 0.85 μg/g in canned tuna fish [27]; and between 0.0 and 0.78 μg/g in canned fish samples [34]. 

Metals such as Cd, Al and Pb, which may pose certain risks to consumers’ health, were also analyzed. Cd and Pb are among the most dangerous toxic substances and are a part of the Priority List of Hazardous Substances according to ATSDR [43]. 

The concentration of Pb in all 32 samples of various canned fish products presented values below the allowed limit (0.30 mg kg^−1^ of f.w.) and was in accordance with European Commission Regulation No. 1881/2006 [36]. The highest value (0.14 mg/kg ww.) was found in the fillets of Atlantic bonito and bluefish with olive oil. Pb values were reported in the literature as 0.0128 ± 0.014 and 0.0183 ± 0.024 in canned tuna and in fresh *Thunnus albacares*, respectively, during a six-year study in Italy [11]; 0.188 mg/kg in canned anchovy and 0.167 mg/kg in canned rainbow trout purchased in Turkey [12]; and between 0.013 and 1.97 μg/g in canned sardines from Saudi Arabia [42].

The Cd contamination level of the samples was within the EU Commission Regulation No. 1881/2006 [36], with a maximum limit of 0.1 mg/kg f.w. Miedico et al. [11] found a Cd content of 0.0295 ± 0.0415 mg/kg f.w for canned tuna, 0.0132 ± 0.0119 mg/kg f.w for fresh *Thunnus albacare*, and 0.0247 ± 0.0108 mg/kg f.w for fresh *Katsuwonus pelamis*, while one sample of unspecified fresh tuna showed a concentration of 0.114 mg/kg, which is over the allowed limit. Winiarska-Mieczan et al. [44] examined several canned fish products available on the Polish market with values of Cd between 0.2 μg/100 g (sprat in tomato sauce) and 1.7 μg/100 g (tuna in olive oil). The average content of Cd in canned rainbow trout on the Turkish market was determined as 0.001 mg/kg, and it was determined as 0.019 mg/kg in canned anchovies [12]. 

Al was detected in only one sample, with a value of 0.79 mg/kg ww., and this is below the permissible limits set by FAO/WHO (2017) [45] at 60 mg/day. This value is similar to or lower than the aluminum concentration found in 102 canned tuna samples in Lebanon (4.756 μg/g) [8]; in canned tuna commercialized in Canada (3.161 μg/g) [46]; and those obtained by Korfali and Hamdan [47] (0.81 mg/kg for eight canned tuna samples collected from the Lebanese market).

### 3.2. Fatty Acids Composition of Canned Fish Samples

The fatty acid composition (expressed as a percentage of the total fatty acid methyl esters) of canned fish products is shown in Table 3. Twenty-five fatty acids (C12:0 to C22:6n-3) were identified. The main fatty acids found in the analyzed canned fish products were oleic (C18:1n-9c, from 15.42 ± 0.63 to 31.05 ± 0.62%), palmitic (C16:0, from 12.85 ± 0.17 to 21.49 ± 0.54%), linoleic (C18:2n-6, from 3.48 ± 0.38 to 20.83 ± 0.85%) and docosahexaenoic acid (C22:6n-3, from 7.61 ± 0.40 to 18.13 ± 0.55%). 

Regarding the groups of fatty acids, the sum of the fatty acid levels decreased in the order of saturated (SFA) > monounsaturated (MUFA) > polyunsaturated (PUFA) in Atlantic bonito in brine (S1) and in bluefish in extra virgin olive oil (S4) and in vegetable oil (S7). PUFAs were the most abundant group only in Atlantic bonito canned in bio extra virgin olive oil (S2), while MUFAs were the most abundant group in bluefish with honey and sunflower oil (S3). In S5 (Atlantic bonito in vegetable oil), the sum of the fatty acid levels followed the pattern SFA > MUFA = PUFA, and in S6 (European sprat in vegetable oil), SFA = MUFA > PUFA. According to previous studies [48], the packing medium had no effect on the content of SFA in canned mackerel samples. However, differences were observed in the total MUFA and total PUFA contents.

The potential health benefits of canned fish products could be evaluated through lipid quality ratios and indices like n-6/n-3, PUFA/SFA, AI, TI, and h/H (Table 4). In our study, PUFA/SFA ratios ranged from 0.58 in bluefish in vegetable oil to 1.26 in Atlantic bonito in bio extra virgin olive oil. Regarding the n-6/n-3 ratios, the analysis showed the lowest value (0.25) in Atlantic bonito in brine and the highest value (1.7) in Atlantic bonito in bio extra virgin olive oil.

Concerning the three nutritional indices, the results for AI and TI were below 1.00 and those for h/H were > 1.00 in all of the studied samples. The lowest AI and TI values and the highest h/H value were detected in Atlantic bonito preserved in bio extra virgin olive oil.

The sum of eicosapentaenoic (EPA) and docosahexaenoic acids (DHA) varied from 11.31 ± 0.57% in bluefish with honey and sunflower oil to 23.13 ± 0.71% in Atlantic bonito preserved in brine, with DHA being the more prominent contributor. According to the European Food Safety Authority (EFSA, 2012), the dietary recommendations for EPA + DHA intake among adults are between 250 and 500 mg/day. Among the studied canned fish products, only bluefish packed in extra virgin olive oil presented an EPA + DHA content (125.39 ± 6.53 mg per 100 g edible portion) lower than the recommended levels. The highest values for EPA + DHA presence (990.27 ± 30.52 mg per 100 g) were detected in the product without any oily filling medium, Atlantic bonito in brine.

### 3.3. Fat-Soluble Vitamins, Antioxidant Pigments and Cholesterol Content

All analyzed canned fish contained significant amounts of fat-soluble vitamins and cholesterol and low levels of astaxanthin and β-carotene. The results are presented in Table 5. The fat-soluble vitamin, antioxidant pigment and cholesterol contents are expressed as an average and standard deviation (mean ± SD). The data are shown as micrograms per 100 g wet weight (μg/100 g ww) for vitamin A and D_3_, astaxanthin and β-carotene, and as milligrams per 100 g wet weight (mg/100 g ww) for cholesterol and vitamin E. 

The comparison of the obtained results with those of our previous studies on the raw tissue of the investigated fish species show convergence in the case of bluefish, but differences in the cases of bonito and sprat. This convergence is attributed to the fact that canning in vegetable oil is considered one of the milder techniques for preparing foods for consumption [51].

The results presented in Table 5 show differences in the amounts of all analytes in the same fish species tissues but with different types of preservation (sunflower or olive oil, brine and olive oil with lemon and honey). The highest and lowest values for vitamin A were reported in the samples of Atlantic bonito in olive oil (S2) and sunflower oil (S5), as well as for bluefish in honey and lemon (S3), at about 200 μg/100 g ww. The Spanish database [52] presented a vitamin A content in bluefish fillets of about 161 μg per 117 g, which is lower compared to our results. On the other hand, the food composition databank (Germany) [53] indicates a higher amount of vitamins A, D3 and E in canned sprat (390 μg·100^−1^ g ww, 14 μg·100^−1^ g ww, 2.31 mg·100^−1^ g ww). These differences are considered as natural, as commented on in our and other authors’ scientific publications. The content of vitamins in fish tissues depends on many factors, such as the biometric characteristics of specimens, the season of the catch, the cooking techniques used, etc. [2,5,54].

Astaxanthin and beta-carotene (β-carotene) are natural antioxidants. Astaxanthin was found to be better at protecting lipid components and phospholipid’s membranes from peroxidation [55]. The literature data regarding the vitamins and antioxidant pigment content of canned fish are very scarce and incomplete. Cholesterol is a component of lipids, and in the body is a precursor of a number of important compounds, including vitamin D_3_ and some hormones, as well as being included in the structure of cell membranes. Its antioxidative activity is 10 and 100 times greater than that of β-carotene and vitamin E, respectively [55].

The quantitative analysis of antioxidant pigments in the samples showed that β-carotene was quantified in all of them, while astaxanthin in bluefish (S7) was below the limit of detection. The data for both pigments (Table 5) varied considerably among the canned fish products, as astaxanthin levels are the highest in Atlantic bonito (S2, 27.3 μg·100^−1^ g ww), sprat (S6, 27.1 μg·100^−1^ g ww) and bluefish (S3, 20.2 μg·100^−1^ g ww), while β-carotene levels are the highest in Atlantic bonito (S1, 7.2 μg·100^−1^ g ww). Data on similar analyses were not found in the scientific literature.

The American Heart Association (AHA) stated a maximum daily cholesterol intake of up to 300 mg, and for individuals at high risk of heart disease, the amount should not be higher than 200 mg [56]. The amounts of cholesterol in canned fish in this study vary within a much narrower range than the other analyzed components (82.7–195.3 mg·100^−1^ g ww). These quantities are consistent with those indicated by other authors in the scientific literature. Romero et al. [57] demonstrated that the range of cholesterol content in canned jurel, sardine, salmon and tuna is between 41 and 86 mg·100^−1^ g ww; Manthey-Karl et al. [58] showed similar data for bonito fillets in brine—51.6 mg·100^−1^ g ww. Additionally, the United States Department of Agriculture (USDA) reports in its database a cholesterol content of sprats in oil samples of 107 mg, which is very close to the value presented in Table 5 (106.7 mg·100^−1^ g ww) [59].

The Bulgarian Ministry of Health, in its Ordinance No. 1/22.01.2018 [60], sets the recommended daily intakes (RDI) for various nutrients, including fat-soluble vitamins. Based on these recommendations, the amounts of vitamins A, D_3_ and E found in the analyzed samples were recalculated as a percentage of the RDI, presented in Table 5. The indicated results show that most of the analyzed canned fish supply significant amounts of the three vitamins, with the highest contribution being for vitamin D_3_. The samples that provide the highest percentage of vitamin A per 100 g serving are S2, S3 and S5 (RDI about 27%); for vitamin D_3_, these are sample S5 (142% RDI) and sample S1 (56% RDI); and for vitamin E, these are samples S1 and S2 with 31.8% and 35.8% RDI, respectively.

### 3.4. Human Health and Food Safety

Besides the numerous benefits of consuming fish and fish products [1,6,7,61], humans are exposed to certain toxic elements via consumption of them. Fishes may accumulate non-essential metals (Pb, Cd and Ni), and therefore can be considered as a public health threat [20,29,33,62,63]. When the risk of toxicity is resolved, analysis of the concentration of the metal in the muscle tissue and the amount of consumed fish is undertaken. 

The results of this study indicate that the EWI values of all the elements measured in the canned fish products were lower than the PTWI limits set by various health organizations (Table 6).

The Joint FAO/WHO Expert Committee on Food Additives (JECFA) in 2013 adopted an updated PTWI value for Cd of 0.025 mg/kg body weight per week [64]. If we assume an average consumption of 190 g of canned fish per week, the exposure to Cd is in the range of 1.9 μg to 5.7 μg for canned bluefish. Herrera-Herrera et al. (2019) [63] reported that EDI/EWI for Cd values in canned tuna and sardine were similar to results of the current study.

Pb is another toxic element related to serious health concerns. The EFSA [65] set a maximum permissible consumption level of Pb of 1.75 (provisionally) mg/kg of body weight per week. It should be noted that in this study, a maximum value of 0.4 μg was calculated (0.2–0.4 μg). Miedico et al. [11] reported a value of 2.6 μg for canned tuna and 3.7 μg for fresh yellowfin tuna for exposure to Pb at an average consumption of 200 mg of canned fish per week.

Ni contamination in the body may be responsible for negative health issues such as lung fibrosis, kidney and cardiovascular diseases, and may lead to respiratory tract cancer [66]. In 2005, EFSA’s Scientific Panel on Contaminants in the Food Chain (CONTAM) adopted an opinion on the risks to public health associated with Ni in food and drinking water, in which it established a tolerable daily intake (TDI) of 2.8 μg/kg Ni/kg bw per day (0.0028 mg/kg Ni/kg b.w or 0.0196 mg/kg Ni/kg b.w tolerable weekly intake) [67]. In this study, the values of Ni were between 0.0021 mg/kg for canned Atlantic Bonito and 0.0124 mg/kg for marinated bluefish. 

The calculated values for the EWI and PTWI showed that the consumption of the canned fish samples was considered safe with regard to Cu, Fe, Zn, Cr, Mn and Al. In addition, many studies have reported that the EDI/EWI values in canned fish are acceptable [8,9,11,62,63,68]. Additionally, there are few studies on EDI or EWI in various canned fish in the literature, which stated that the EDI/EWI values for Fe, Cu, Zn, Al, Cr, Cd, Mn, Pb and Ni are under the PTWI levels stated by health organizations [69], so these fish samples do not pose risks to consumers’ health.

**Table 6 foods-13-00936-t006:** Estimated weekly intake (EWI; mg/kg BW) values for each element analyzed in various canned fish species.

		Cu	Fe	Zn	Cr	Mn	Pb	Cd	Ni	Al
Estimated weekly intake (mg/week/70 kg body weight)	S1	nd	0.0586	0.0062	nd	nd	nd	0.0000	0.0000	0.1501
	S2	0.00043	0.0524	0.0055	nd	nd	nd	0.0057	0.0021	0.0000
	S3	0.00027	0.0537	0.0074	0.0014	0.0005	0.0005	0.0019	0.0084	0.0000
	S4	0.00041	0.0679	0.0092	0.0010	0.0005	0.0005	0.0048	0.0072	0.0000
	S5	nd	0.0603	0.0064	0.0015	0.0002	0.0002	0.0048	0.0000	0.0000
	S6	0.00043	0.1186	0.0208	0.0008	0.0006	0.0006	0.0048	0.0086	0.0000
	S7	0.00130	0.0556	0.0059	nd	0.0004	0.0004	0.0029	0.0124	0.0000
PTWI (mg/kg body weight)		3.5 [70]	0.0056 [70]	7 [70]	0.7 [71]	-	0.025 [72]	0.007 [64]	0.0028 (TDI) [73]	2 [70]
PTWI (70 kg body weight)		245	0.392	490	49	-	1.75	0.49	-	140

PTWI = provisional permissible tolerable weekly intake; TDI = tolerable daily intake.

The values of the hazard quotient for the benefit–risk ratio, HQEFA, are presented in Table 7. The calculated values ranged from 0.001 (Al) for Atlantic bonito in brine to 1.405 (Cr) for bluefish in extra virgin olive oil. In most canned fish, the HQ_EFA_ values were below 1, therefore posing no risk to people when consuming these products. However, in bluefish preserved in extra virgin olive oil, HQ_EFA_ for Cr > 1, which means that the risk of consumption of this product may outweigh the benefits of PUFA intake. 

### 3.5. Importance of Fish Consumption for the Population of Bulgaria

Corresponding to the WHO [71], the low consumption of fish in Bulgaria is associated with a risk of cardiovascular diseases in different age groups. This may be due to the low intake of omega-3 PUFAs, fat-soluble vitamins such as vitamin D3, and essential microelements. One Bulgarian regulatory document [60] related to health encouraged fish consumption, including canned fish. Regardless of these recommendations, sufficient adequate information has not been provided on the content of important biologically active compounds in canned fish species traditionally consumed in Bulgaria. There is no information on their microelement, fatty acid and fat-soluble vitamin contents. This makes it difficult for nutritionists and concerned consumers to make informed choices among available local canned fish species. The current study represents novel information on the nutritional value and safety of canned fish species sold in Bulgaria. Canned products contain a diverse fatty acid, vitamin and carotenoid profile. Significant amounts of essential microelements such as iron and zinc were found, while potentially toxic elements were found in safe amounts. The main significance of this study is in emphasizing the amounts of EPA+DHA, vitamin D3 and cholesterol.

## 4. Conclusions

The results of the present study highlight the benefits and risks of consuming different local canned fish products from Bulgaria. Although fish consumption in Bulgaria is lower compared to that in other countries, it is still important to regularly test locally produced canned fish for the presence of elements of toxicological concern and essential elements to protect consumer health. Additionally, the analysis of fatty acid composition and the contents of fat-soluble vitamins, antioxidant pigments and cholesterol provide us with a full base for an assessment of the overall adverse effects on human health.

The data related to the essential and toxic elements are within the maximum acceptable limits set by different health organizations or within the literature data. All canned fish samples presented favorable n-6/n-3 ratios and AI, TI and h/H indices. Considering the EPA and DHA, Atlantic bonito preserved in brine and bluefish preserved in honey and sunflower oil and in vegetable oil appear as the most beneficial canned products for the human diet, while bluefish packed in extra virgin olive oil is the least advantageous. 

In accordance with the RDI for Bulgaria, Atlantic bonito in sunflower oil is characterized as a very good source of vitamin A and D_3_ per 100 g serving. The cholesterol content established in all analyzed canned fish samples characterizes them as a healthy food with safe levels of this nutrient.

Although the EWI levels in the seven studied canned fish products were not found to pose a risk, the sample of bluefish preserved in extra virgin olive oil showed a HQ_EFA_ for Cr > 1, which might be associated with a risk in which consumption of this product may outweigh the benefits of PUFA intake. The regular monitoring of the hazard quotients for canned fish is highly advisable in order to protect people from the adverse effects of canned fish consumption and to provide healthy diets at the same time.

Moreover, further research should be carried out to provide up-to-date information for benefit–risk assessment of the consumption of various canned fish species. As Usydes et al. [74] stated in their research, consumers should pick from fish and fish products available on the market in such a way so they can maximize the benefits (high PUFA and vitamin content) and minimize the risk (low toxic content). 

## Figures and Tables

**Table 1 foods-13-00936-t001:** Information for canned fish products. Ingredients are listed in the order given on the label.

Sample	Fish Type	Packaging	Percent of Fish	Weigh Total/Net Content	N	Additives and Comments
S1	Atlantic bonito	Metal can	70%	115 g/81 g	6	Filets, boneless, water, salt
S2	Atlantic bonito	Metal can	70%	115 g/81 g	6	Filets, boneless, bio extra virgin olive oil, salt
S3	Bluefish	Glass jar	63%	220 g/140 g	3	Filets, boneless, sea salt, citric acid, honey (<0.5%), sunflower oil, lemon
S4	Bluefish	Glass jar	62%	220 g/140 g	3	Filets, boneless, sea salt, extra virgin olive oil, citric acid
S5	Atlantic bonito	Glass jar	55%	180 g/100 g	3	Filets, boneless, water, sea salt, vegetable oil
S6	European sprat	Glass jar	47%	180 g/100 g	4	Whole fish, water, sea salt, vegetable oil, lemon, dill
S7	Bluefish	Metal can	67%	115 g/80 g	5	Filets, boneless, water, sea salt, vegetable oil

N—number of composite samples.

**Table 2 foods-13-00936-t002:** Concentration of essential elements and elements of toxicological concern in canned fish (in mg/kg ww; mean ± SD).

	Cu	Fe	Zn	Cr	Mn	Pb	Cd	Ni	Al
S1	nd	21.61 ± 0.21 ^c^	2.29 ± 0.08 ^c^	nd	nd	0.14 ± 0.09 ^a^	nd	nd	0.79 ± 0.05
S2	0.16 ± 0.08 ^b^	19.34 ± 0.14 ^e^	2.02 ± 0.06 ^c^	nd	nd	0.01 ± 0.01 ^a^	0.032 ± 0.011 ^a^	0.011 ± 0.01 ^a^	nd
S3	0.10 ± 0.03 ^b^	19.79 ± 0.15 ^de^	2.74 ± 0.10 ^bc^	0.51 ± 0.18 ^a^	0.17 ± 0.02 ^ab^	0.08 ± 0.05 ^a^	0.011 ± 0.010 ^a^	0.044 ± 0.01 1^a^	nd
S4	0.15 ± 0.07 ^b^	25.02 ± 0.37 ^b^	3.40 ± 0.35 ^b^	0.37 ± 0.13 ^a^	0.17 ± 0.04 ^ab^	0.14 ± 0.05 ^a^	0.025 ± 0.001 ^a^	0.038 ± 0.032 ^a^	nd
S5	nd	22.17 ± 0.39 ^c^	2.36 ± 0.42 ^bc^	0.54 ± 0.11 ^a^	0.08 ± 0.01 ^c^	0.03 ± 0.03 ^a^	0.025 ± 0.001 ^a^	nd	nd
S6	0.16 ± 0.01 ^b^	43.69 ± 0.39 ^a^	7.65 ± 0.79 ^a^	0.29 ± 0.2 ^a^	0.23 ± 0.02 ^a^	0.09 ± 0.05 ^a^	0.025 ± 0.001 ^a^	0.045 ± 0.028 ^a^	nd
S7	0.48 ± 0.1 ^a^	20.54 ± 0.37 ^d^	2.18 ± 0.29 ^c^	nd	0.13 ± 0.02 ^bc^	0.05 ± 0.03 ^a^	0.015 ± 0.014 ^a^	0.065 ± 0.022 ^a^	nd

nd—not detected; values in a column without common superscript are significantly different (*p* < 0.05).

**Table 3 foods-13-00936-t003:** Fatty acid profiles of canned fish (expressed as % of total fatty acids; mean ± SD).

	S1	S2	S3	S4	S5	S6	S7
C12:0	nd	1.08 ± 0.16	nd	nd	nd	nd	1.22 ± 0.03
C13:0	0.67 ± 0.06	nd	nd	nd	nd	nd	0.72 ± 0.02
C14:0	3.73 ± 0.26	2.67 ± 0.12	3.97 ± 0.07	4.42 ± 0.30	4.03 ± 0.13	6.44 ± 0.32	4.33 ± 0.09
C15:0	1.18 ± 0.08	0.95 ± 0.09	1.09 ± 0.06	1.15 ± 0.08	1.33 ± 0.05	1.16 ± 0.07	1.26 ± 0.03
C16:0	16.71 ± 3.09	12.85 ± 0.17	14.76 ± 0.42	16.44 ± 1.24	20.79 ± 0.37	14.76 ± 0.13	21.49 ± 0.54
C17:0	1.44 ± 0.08	1.09 ± 0.10	1.18 ± 0.04	1.33 ± 0.10	1.47 ± 0.06	1.15 ± 0.08	1.39 ± 0.01
C18:0	7.25 ± 0.41	5.50 ± 0.21	5.18 ± 0.08	5.70 ± 0.58	6.75 ± 0.29	3.25 ± 0.18	6.65 ± 0.29
C20:0	1.65 ± 0.14	1.56 ± 0.18	1.67 ± 0.10	1.95 ± 0.14	1.89 ± 0.10	1.98 ± 0.14	1.79 ± 0.03
C21:0	0.80 ± 0.07	1.30 ± 0.13	0.77 ± 0.05	0.84 ± 0.06	0.87 ± 0.05	0.97 ± 0.07	0.85 ± 0.02
C22:0	1.52 ± 0.13	1.57 ± 0.17	1.61 ± 0.10	1.71 ± 0.13	1.73 ± 0.09	1.93 ± 0.14	1.68 ± 0.03
C23:0	0.75 ± 0.06	nd	nd	0.78 ± 0.06	0.81 ± 0.05	nd	0.80 ± 0.02
C24:0	1.66 ± 0.14	1.60 ± 0.20	1.63 ± 0.12	1.77 ± 0.13	1.83 ± 0.10	2.02 ± 0.16	1.78 ± 0.04
SFA	37.36 ± 1.72 ^b^	30.17 ± 1.05 ^c^	31.86 ± 0.33 ^c^	36.10 ± 2.78 ^b^	41.49 ± 0.38 ^a^	33.65 ± 1.06 ^bc^	43.96 ± 0.80 ^a^
C14:1	nd	0.83 ± 0.12	0.84 ± 0.07	0.92 ± 0.07	0.98 ± 0.05	nd	0.94 ± 0.01
C16:1	4.14 ± 0.16	2.39 ± 0.05	3.94 ± 0.11	4.36 ± 0.48	3.64 ± 0.34	3.43 ± 0.07	4.19 ± 0.01
C18:1n-9t	0.81 ± 0.08	0.78 ± 0.10	0.81 ± 0.05	0.87 ± 0.07	0.89 ± 0.06	nd	0.87 ± 0.02
C18:1n-9c	23.57 ± 1.20	25.66 ± 1.79	31.05 ± 0.62	18.53 ± 1.85	18.44 ± 0.43	15.42 ± 0.63	18.52 ± 1.12
C20:1	2.06 ± 0.18	1.30 ± 0.13	1.77 ± 0.08	6.88 ± 8.60	2.36 ± 0.06	12.25 ± 0.88	2.63 ± 0.15
C22:1	nd	nd	1.26 ± 0.31	1.33 ± 0.41	1.67 ± 0.06	1.10 ± 0.08	1.76 ± 0.75
C24:1	1.25 ± 0.08	0.95 ± 0.08	1.14 ± 0.14	1.26 ± 0.11	1.43 ± 0.10	1.51 ± 0.11	1.50 ± 0.05
MUFA	31.82 ± 1.55 ^b^	31.91 ± 1.33 ^b^	40.80 ± 0.59 ^a^	34.15 ± 5.66 ^b^	29.41 ± 0.60 ^b^	33.70 ± 1.14 ^b^	30.40 ± 1.08 ^b^
C18:2n-6c	3.48 ± 0.38	20.83 ± 0.85	12.69 ± 0.78	10.68 ± 1.79	3.88 ± 0.09	14.63 ± 2.20	3.80 ± 0.46
C18:2n-6t	nd	nd	Nd	nd	nd	nd	nd
C18:3n-6	nd	0.85 ± 0.11	Nd	nd	nd	nd	nd
C18:3n-3	1.60 ± 0.18	1.47 ± 0.14	1.38 ± 0.13	1.47 ± 0.18	1.53 ± 0.03	nd	1.45 ± 0.01
C20:2	1.01 ± 0.08	0.93 ± 0.10	0.92 ± 0.07	1.00 ± 0.07	1.03 ± 0.06	1.13 ± 0.08	1.01 ± 0.03
C20:3n-6	nd	nd	nd	nd	nd	nd	nd
C20:3n-3	nd	nd	nd	nd	nd	nd	nd
C20:4n-6 ARA	1.60 ± 0.11	1.25 ± 0.11	1.04 ± 0.06	1.18 ± 0.09	1.25 ± 0.06	1.23 ± 0.09	1.19 ± 0.02
C22:2	nd	nd	nd	nd	nd	nd	nd
C20:5n-3 EPA	5.00 ± 0.49	3.23 ± 0.12	3.70 ± 0.18	4.13 ± 0.38	4.95 ± 0.08	6.59 ± 0.15	4.15 ± 0.03
C22:6n-3 DHA	18.13 ± 0.55	9.35 ± 0.32	7.61 ± 0.40	11.29 ± 0.44	16.46 ± 0.94	9.08 ± 0.25	14.03 ± 0.60
PUFA	30.83 ± 0.49 ^bc^	37.92 ± 0.78 ^a^	27.34 ± 0.38 ^cd^	29.75 ± 2.89 ^bc^	29.10 ± 0.89 ^bcd^	32.65 ± 1.95 ^b^	25.64 ± 0.33 ^d^

Results represent mean values ± standard deviation (*n* = 3); SFA: saturated fatty acids; MUFA: monounsaturated fatty acids; PUFA: polyunsaturated fatty acids; *nd*: not detected; values in a row without common superscript are significantly different (*p* < 0.05).

**Table 4 foods-13-00936-t004:** Nutritional quality indexes of canned fish.

	S1	S2	S3	S4	S5	S6	S7
n-3	24.73 ± 0.60 ^a^	14.06 ± 0.26 ^de^	12.69 ± 0.51 ^e^	16.89 ± 0.96 ^c^	22.94 ± 0.99 ^a^	15.66 ± 0.15 ^cd^	19.64 ± 0.57 ^b^
n-6	6.10 ± 0.56 ^d^	23.86 ± 0.96 ^a^	14.65 ± 0.89 ^bc^	12.85 ± 1.94 ^c^	6.16 ± 0.19 ^d^	16.98 ± 2.09 ^b^	6.00 ± 0.42 ^d^
n-6/n-3	0.25 ± 0.03 ^d^	1.70 ± 0.09 ^a^	1.16 ± 0.11 ^b^	0.76 ± 0.08 ^c^	0.27 ± 0.02 ^d^	1.09 ± 0.14 ^b^	0.31 ± 0.03 ^d^
PUFA/SFA	0.83 ± 0.05 ^c^	1.26 ± 0.05 ^a^	0.86 ± 0.01 ^bc^	0.82 ± 0.02 ^cd^	0.70 ± 0.03 ^de^	0.97 ± 0.09 ^b^	0.58 ± 0.01 ^e^
DHA/EPA	3.65 ± 0.39 ^a^	2.89 ± 0.10 ^b^	2.06 ± 0.04 ^c^	2.74 ± 0.16 ^b^	3.32 ± 0.14 ^a^	1.38 ± 0.07 ^d^	3.38 ± 0.17 ^a^
DHA + EPA	23.13 ± 0.71 ^a^	12.58 ± 0.40 ^d^	11.31 ± 0.57 ^d^	15.43 ± 0.80 ^c^	21.41 ± 1.02 ^a^	15.66 ± 0.15 ^c^	18.19 ± 0.57 ^b^
AI *	0.51 ± 0.05 ^c^	0.35 ± 0.01 ^d^	0.45 ± 0.01 ^c^	0.54 ± 0.06 ^bc^	0.63 ± 0.01 ^ab^	0.61 ± 0.03 ^b^	0.71 ± 0.03 ^a^
TI **	0.28 ± 0.02 ^d^	0.30 ± 0.01 ^cd^	0.36 ± 0.01 ^ab^	0.35 ± 0.02 ^b^	0.35 ± 0.01 ^b^	0.33 ± 0.01 ^bc^	0.40 ± 0.02 ^a^
h/H ***	2.65 ± 0.41 ^bc^	3.98 ± 0.11 ^a^	3.07 ± 0.10 ^b^	2.26 ± 0.06 ^cd^	1.87 ± 0.04 ^de^	2.22 ± 0.14 ^cd^	1.67 ± 0.06 ^e^
DHA + EPA, mg/100 g	990.27 ± 30.52 ^a^	374.13 ± 11.81 ^d^	718.70 ± 35.97 ^b^	125.39 ± 6.53^e^	484.77 ± 23.18 ^c^	349.79 ± 3.26 ^d^	691.53 ± 21.83 ^b^

Results represent mean values ± standard deviation (*n* = 3); TI: thrombogenicity index; AI: atherogenicity index [49]; h/H: hypocholesterolemic to hypercholesterolemic ratio [50]; * AI = [C12:0 + (4 × C14:0) + C16:0]/(n-6PUFA + n-3PUFA + MUFA); ** TI = (C14:0 + C16:0 + C18:0)/[(0.5MUFA) + (0.5n-6PUFA) + (3n-3PUFA) + (n-3PUFA/n-6PUFA)]; *** h/H = (C18:1n-9 + C18:2n-6 + C18:3n-3 + C20:4n-6 + C20:5n-3 + C22:6n-3)/(C14:0 + C16:0). Values in a row without common superscript are significantly different (*p* < 0.05).

**Table 5 foods-13-00936-t005:** Fat-soluble vitamin, antioxidant pigment and cholesterol contents in canned fish (mean ± SD) and the percentage of the relative daily intake of vitamins.

	S1	S2	S3	S4	S5	S6	S7
Vitamin A, μg·100 g^−1^ ww	125.2 ± 9.7 ^b^	207.4 ± 21.3 ^a^	208.4 ± 20.3 ^a^	129.3 ± 14.3 ^b^	200.7 ± 19.3 ^a^	13.03 ± 0.6 ^d^	67.7 ± 9.1 ^c^
Vitamin D_3_, μg·100 g^−1^ ww	8.4 ± 0.8 ^c^	3.2 ± 0.4 ^d^	2.9 ± 0.5 ^d^	9.5 ± 1.4 ^b^	21.3 ± 2.4 ^a^	4.1 ± 0.4 ^d^	4.5 ± 0.4 ^d^
Vitamin E, mg·100 g^−1^ ww	4.3 ± 0.6 ^a^	4.8 ± 0.4 ^a^	2.58 ± 0.15 ^b^	2.14 ± 0.2 ^b^	0.6 ± 0.07 ^c^	0.35 ± 0.04 ^c^	0.16 ± 0.02 ^c^
Astaxanthin, μg·100 g^−1^ ww	4.3 ± 0.4 ^b^	27.3 ± 4.3 ^a^	4.1 ± 0.2 ^b^	20.2 ± 4.3 ^a^	4.3 ± 0.1 ^c^	27.1 ± 4.3 ^a^	nd
β-Carotene, mg·100 g^−1^ ww	7.2 ± 0.7 ^a^	0.6 ± 0.03 ^c^	1.7 ± 0.2 ^b^	1.1 ± 0.4 ^bc^	1.73 ± 0.07 ^b^	0.36 ± 0.05 ^c^	0.53 ± 0.03 ^c^
Cholesterol, mg·100 g^−1^ ww	151.2 ± 17.4 ^b^	107.1 ± 12.8 ^c^	195.3 ± 20.6 ^a^	153.8 ± 13.8 ^b^	170 ± 2.8 ^ab^	106.7 ± 12.8 ^c^	82.7 ± 9.1 ^c^
%RDI Vit A	16.7	27.6	27.7	17.2	26.7	1.7	9.0
%RDI Vit D_3_	56	21.3	19.3	63.3	142	27.3	30
%RDI Vit E	31.8	35.8	19.1	15.9	4.4	2.6	1.2

nd—not detected; values in a row without common superscript are significantly different (*p* < 0.05).

**Table 7 foods-13-00936-t007:** Hazard quotients, HQ_EFA_, for the benefit–risk ratio of essential fatty acids vs. metals for the consumption of canned fish products.

Sample	S1	S2	S3	S4	S5	S6	S7
Cu	-	0.015	0.005	0.043	-	0.016	0.025
Fe	0.045	0.105	0.056	0.407	0.093	0.255	0.060
Zn	0.011	0.026	0.018	0.129	0.023	0.104	0.015
Cr	-	-	0.338	1.405	0.530	0.395	-
Mn	-	-	0.068	0.387	0.047	0.188	0.054
Pb	0.101	0.019	0.080	0.798	0.044	0.184	0.052
Cd	-	0.115	0.020	0.285	0.074	0.102	0.310
Ni	-	0.002	0.004	0.022	-	0.009	0.007
Al	0.001	-	-	-	-	-	-

## Data Availability

The original contributions presented in the study are included in the article, further inquiries can be directed to the corresponding author.

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
