# Peer review of "Metal Content, Fatty Acid and Vitamins in Commercially Available Canned Fish on the Bulgarian Market: Benefit–Risk Ratio Intake"

_foods, 2024, doi:10.3390/foods13060936_

Round 1
Reviewer 1 Report
Comments and Suggestions for Authors
The present manuscript aims to evaluate some metals of toxicological concern (Al, Cd, Cr, Cu, Fe, Mn, Ni, Pb and Zn) but also fatty acid composition, fat soluble vitamins and antioxidant pigments content of various canned fish purchased from the Bulgarian market. The estimated weekly intake and benefit-risk ratio for human health based on trace elements and n-3 LC-PUFAs contents in canned fishes were assessed.
The paper is well organized and the results are well presented. The topic is of current interest and may contribute to the enrichment of the specialized literature, but the novelty of the topic addressed should be better highlighted. I suggest to slightly modify the title to emphasize better the importance of the approaching the theme.
Table 1 - I recommend including some horizontal lines to separate the samples and thus the information presented to be more evident.
In column 1 (Sample), some samples have the ‘dot’ after codification.
R95: Replace ‘mg/l’ with ‘mg/L’. Check in all the paper.
Table 3: The columns are not well aligned in the direction of the text and reading is difficult.
R343: ‘23.13 ± 0.71’ include the unit measure
R397: ‘27.1μg.100−1 g ww’ Add space between value and unit measure; check all the text.
R405: Replace ‘Romero et al. (1996) [57]’ with ‘Romero et al. [57]’
R407: Replace ‘Manthey-Karl et al. (2014) [58]’ with ‘Manthey-Karl et al. [58].
R439: Replace ‘Miedico et al. (2020) [11]’ with ‘Miedico et al. [11]’
R489: ‘The main interest is focused on the amounts of EPA+DHA, vitamin D3 and cholesterol’ - add a dot at the end of the phrase.
Table 4: I recommend that all tables have the same presentation, "mean±SD" and not in different columns.
I recommend presenting the key results in a more attractive way (charts, graphs) to improve the appearance of the article.
Conclusions: Provide recommendations for consumers based on your analysis. You can suggest some future prospects to further enhance understanding.
Comments on the Quality of English Language
Moderate editing of English language required.
Author Response
Dear Editor (s),
Thank you for very valuable and useful comments of respectful reviewers’ on our manuscript entitled: “ Metal level, fatty acid and vitamins in commercially available canned fish on the Bulgarian market. Benefit-risk ratio intake " by K.Peycheva et al.
In the revised copy the following points were considered:
Comments to the Author:
The present manuscript aims to evaluate some metals of toxicological concern (Al, Cd, Cr, Cu, Fe, Mn, Ni, Pb and Zn) but also fatty acid composition, fat soluble vitamins and antioxidant pigments content of various canned fish purchased from the Bulgarian market. The estimated weekly intake and benefit-risk ratio for human health based on trace elements and n-3 LC-PUFAs contents in canned fishes were assessed.
The paper is well organized, and the results are well presented. The topic is of current interest and may contribute to the enrichment of specialized literature, but the novelty of the topic addressed should be better highlighted. I suggest slightly modifying the title to emphasize better the importance of approaching the theme.
Table 1 - I recommend including some horizontal lines to separate the samples and thus the information presented to be more evident.
It was corrected in edited manuscript
In column 1 (Sample), some samples have the ‘dot’ after codification.
It was corrected in edited manuscript
R95: Replace ‘mg/l’ with ‘mg/L’. Check in all the papers.
It was corrected in edited manuscript
Table 3: The columns are not well aligned in the direction of the text and reading is difficult.
It was corrected in edited manuscript
R343: ‘23.13 ± 0.71’ include the unit measure
It was corrected in edited manuscript
R397: ‘27.1μg.100−1 g ww’ Add space between value and unit measure; check all the text.
It was corrected in edited manuscript
R405: Replace ‘Romero et al. (1996) [57]’ with ‘Romero et al. [57]’
It was corrected in edited manuscript
R407: Replace ‘Manthey-Karl et al. (2014) [58]’ with ‘Manthey-Karl et al. [58].
It was corrected in edited manuscript
R439: Replace ‘Miedico et al. (2020) [11]’ with ‘Miedico et al. [11]’
It was corrected in edited manuscript
R489: ‘The main interest is focused on the amounts of EPA+DHA, vitamin D3 and cholesterol’ - add a dot at the end of the phrase.
It was corrected in edited manuscript
Table 4: I recommend that all tables have the same presentation, "mean±SD" and not in different columns.
It was corrected in edited manuscript
I recommend presenting the key results in a more attractive way (charts, graphs) to improve the appearance of the article.
It was corrected in edited manuscript
Conclusions: Provide recommendations for consumers based on your analysis. You can suggest some future prospects to further enhance understanding.
The authors appreciate this valuable comment and slightly modified the Conclusion part in respect with it
Moderate editing of English language required.
It was corrected in edited manuscript
Reviewer 2 Report
Comments and Suggestions for Authors
The content of selected metals, fatty acids and fat-soluble vitamins were determined in canned fish products from the Bulgarian market. Moreover, authors evaluated health risks e.g. benefit-risk ratio intake. This type of research is not a scientific novelty, however an attention should be paid to the numerous analytical methods used during the research (ICP OES, GC-MS, HPLC). For this reason, I am asking to complete the methodologies so that readers can use them in their research. Please refer to the comments below:
1. I am not convinced to use the word "levels" in the case of analytes (metals, acids, vitamins etc.). I recommend using the terms “content” or “value(s)” more often than “levels”.
2. Please unify the significant numbers, e.g. Tables 2, 3.
3. According to latest recommendations, a hyphen “-“ in acronyms of analytical techniques should be kept only for hyphenated techniques e.g. GC-MS, while others should be written with space, e.g. ICP OES. Please make change the whole text.
4. Section 2.1. I do not understand the sampling. You wrote about 5 brands of 30 products while there are 7 samples in Table 1. Why did not you analyse all of 30 products?
5. Lines 79. “A minimum of three samples per brand of canned fish was randomly purchased”. I understand that you used 3 products per samples however you did not distinguish standard deviation between triplicate of the same sample and three different cans. Please improve the description and explain your point of view.
6. Line 82. Please complete the description of the homogenization method.
7. Lines 102-104. Please add more operating parameters, i.e. RF power, acquisition time, emission lines, detection limits.
8. Line 104. Nebulizer gas flow seems to be very low. Please specify which type of nebulizer you used and what sample liquid flow was used to maintain efficient sample nebulization.
9. Lines 143-144. Please add more operating parameters of chromatographic run, e.g. eluent(s) type, eluent(s) flow, sample volume etc., as well as additional equipment (degasser, detection type, column oven etc.).
Author Response
Dear Editor (s),
Thank you for very valuable and useful comments of respectful reviewers’ on our manuscript entitled: “ Metal level, fatty acid and vitamins in commercially available canned fish on the Bulgarian market. Benefit-risk ratio intake " by K.Peycheva et al.
In the revised copy the following points were considered:
Comments to the Author:
The content of selected metals, fatty acids and fat-soluble vitamins were determined in canned fish products from the Bulgarian market. Moreover, authors evaluated health risks e.g. benefit-risk ratio intake. This type of research is not a scientific novelty, however an attention should be paid to the numerous analytical methods used during the research (ICP OES, GC-MS, HPLC). For this reason, I am asking to complete the methodologies so that readers can use them in their research. Please refer to the comments below:
1.I am not convinced to use the word "levels" in the case of analytes (metals, acids, vitamins etc.). I recommend using the terms “content” or “value(s)” more often than “levels”.
Thanks for the valuable comment. It was corrected in edited manuscript.
2. Please unify the significant numbers, e.g. Tables 2, 3.
It was corrected in edited manuscript
3. According to latest recommendations, a hyphen “- “in acronyms of analytical techniques should be kept only for hyphenated techniques e.g. GC-MS, while others should be written with space, e.g. ICP OES. Please make change the whole text.
It was corrected in edited manuscript
4. Section 2.1. I do not understand the sampling. You wrote about 5 brands of 30 products while there are 7 samples in Table 1. Why did not you analyse all of 30 products?
This study involved the procurement of 7 canned fish species (e.g. Atlantic Bonito, Bluefish and etc.) in brine, in olive oil, with spices from five different companies that were obtained from local markets in Bulgaria. The total number of cans/jars were 30 in order to make a homogeneous representative sample for further analytical determination.
5. Lines 79. “A minimum of three samples per brand of canned fish was randomly purchased”. I understand that you used 3 products per samples however you did not distinguish standard deviation between triplicate of the same sample and three different cans. Please improve the description and explain your point of view.
Authors agreed on that remarkable comment and changed it to “A minimum of three samples were selected from each seven canned fish to provide a representative dataset”
6. Line 82. Please complete the description of the homogenization method.
It was corrected in edited manuscript adding an information related the homogenation
7. Lines 102-104. Please add more operating parameters, i.e. RF power, acquisition time, emission lines, detection limits.
The additional information was presented in the revised MS
8. Line 104. Nebulizer gas flow seems to be very low. Please specify which type of nebulizer you used and what sample liquid flow was used to maintain efficient sample nebulization.
A technical error had been occurred, which was eliminated after this valuable comment
9. Lines 143-144. Please add more operating parameters of chromatographic run, e.g. eluent(s) type, eluent(s) flow, sample volume etc., as well as additional equipment (degasser, detection type, column oven etc.).
Chromatography conditions and sample preparation are described in detail by Dobreva et al. (Dobreva, D.A.; Panayotova, V.; Stancheva, R.; Stancheva, M. Simultaneous HPLC determination of fat soluble vitamins, carotenoids and cholesterol in seaweed and mussel tissue. Bulg Chem Commun 2017, 49, 112-117.)
Reviewer 3 Report
Comments and Suggestions for Authors
On manuscript on Peycheva et al on the safety and nutritional value on canned fish spurred on Bulgaria. Although manuscript carries relevant data and analysis on lacks additional clarifications. First on approach on methods comparison already seen on other countries or canned food? On methods why authors choose on assess safety on canned fish on Bulgaria? I mean on this a novel governmental recommendation? On lack on regulations on Bulgaria? On any intoxications? Or on other concerns on canned products on other countries? On safety assessment on methods on standard according on international or national guidelines? On all canned fish analysed collected on Bulgaria? Another remark refers on whole fish analysed or on portions? Also, on nutritional values on compared on metals? I mean findings on safety on canned fish on Bulgaria or on fish canned on Bulgaria or elsewhere?
Finally, on results authors should provide an explanation on the reasons on the safety on their canned fish on comparison on other discussed products.
Author Response
Dear Editor (s),
Thank you for very valuable and useful comments of respectful reviewers’ on our manuscript entitled: “ Metal level, fatty acid and vitamins in commercially available canned fish on the Bulgarian market. Benefit-risk ratio intake " by K.Peycheva et al.
In the revised copy the following points were considered:
Comments to the Author:
On manuscript on Peycheva et al on the safety and nutritional value on canned fish spurred on Bulgaria. Although manuscript carries relevant data and analysis on lacks additional clarifications.
First on approach on methods comparison already seen on other countries or canned food? On methods why authors choose on assess safety on canned fish on Bulgaria? I mean on this a novel governmental recommendation? On lack on regulations on Bulgaria? On any intoxications? Or on other concerns on canned products on other countries? On safety assessment on methods on standard according on international or national guidelines? On all canned fish analysed collected on Bulgaria?
Today, there is a growing awareness about the importance of eating nutritious foods and fish is gaining momentum as a result of its unique nutritional benefits. This MS represents a preliminary study on the topic concerning various canned fish species since such kind of research is not available in our country. Additionally, in order to synchronized our government regulation with the ones set by the EU, the experts in the filed needs a relevant database. The authors laid the foundation and think to expand this topic in the near future. In this aspect the authors tried to compare this primary data with those stated in the literature.
Another remark refers on whole fish analysed or on portions?
A minimum of three samples were selected from each seven canned fish to provide a representative database. Only fish fillets were used. They were homogenized, digested immediately and analyses using the proper analytical approaches.
Also, on nutritional values on compared on metals? I mean findings on safety on canned fish on Bulgaria or on fish canned on Bulgaria or elsewhere?
Unfortunately, there is no information regarding nutritional values of various canned fish species in our country. That is why the authors compare their data with those published in the neighbouring countries, which have similar feeding habits.
Finally, on results authors should provide an explanation on the reasons on the safety on their canned fish on comparison on other discussed products.
The authors appreciate this valuable comment. An additional information was included in the Conclusion sector.